# Five-Mode Tucker-LoRA for Video Diffusion on Conv3D Backbones

## Abstract

Parameter-efficient fine-tuning for text-to-video diffusion remains challenging. Most LoRA-style adapters either flatten 3D kernels into 2D matrices or add temporal-only modules, which breaks the native structure of Conv3D backbones. We present a five-mode Tucker-LoRA that learns a Tucker residual directly on the 5-D convolutional weight update across output/input channels, time, height, and width. This preserves spatio-temporal geometry and enables mode-wise rank budgets; setting some ranks to one (or the temporal rank to zero) recovers common 2D or temporal-only adapters. We instantiate the adapter in VideoCrafter (Conv3D U-Net) and AnimateDiff (2D+motion) under a unified 16×224 evaluation protocol on MSR-VTT. The method achieves a favorable memory–quality trade-off compared with strong 2D/pseudo-3D baselines and reaches target FVD earlier in time-to-target analysis. Results and ablations suggest that respecting the full dimensionality of video kernels is key for budgeted, tensorized adaptation.

## 1 Introduction

Text-to-video diffusion models have recently achieved strong fidelity and temporal coherence, driven by large-scale training and Conv3D (3D UNet) backbones (Chen et al., 2023b; Guo et al., 2023). Adapting these models remains costly: a Conv3D kernel is a five-mode tensor over output/input channels, time, height, and width, yielding large parameter counts and memory footprints.

Parameter-efficient fine-tuning (PEFT) is effective in language and image models. LoRA (Hu et al., 2021) introduces low-rank residual updates and substantially reduces trainable parameters, but common instantiations for video either flatten convolution weights into matrices or target attention/2D components only, misaligning with the native spatio–temporal structure of Conv3D. In practice, VideoCrafter-style models rely on full 3D convolutions without structured low-rank adaptation, while AnimateDiff employs *pseudo-3D* adapters that operate on 2D layers or motion MLPs rather than the full 3D filters (Guo et al., 2023; Chen et al., 2024).

We propose a **five-mode (5D) Tucker-LoRA** that learns a Tucker *residual* directly on the 5-D convolutional weight update across $(O, I, T, H, W)$. This formulation preserves the spatio–temporal geometry of video kernels and enables mode-wise rank control; setting certain ranks to 1 (or the temporal rank t=0) recovers 2D or temporal-only LoRA as special cases. We instantiate the adapter in two representative backbones—VideoCrafter (Conv3D UNet) and AnimateDiff (2D UNet with motion modules)—under a unified training and evaluation pipeline.

Under a unified 16×224 protocol on MSR-VTT, our 5D Tucker-LoRA attains favorable memory–quality trade-offs compared with 2D/pseudo-3D adapters. Time-to-target analysis further indicates earlier attainment of a practical FVD band on the Conv3D backbone, while AnimateDiff retains higher throughput. These results suggest that respecting the full dimensionality of Conv3D kernels is a useful design principle for budgeted, tensorized adaptation in video diffusion.

## 2 Preliminaries: Video Latent Diffusion and 3D Convolutions

**Problem setup.** We consider text–to–video generation where a short video clip $x_0 \in \mathbb{R}^{T \times H \times W \times 3}$ (with $T$ frames) is generated conditioned on a text prompt $y$. Following latent diffusion, a pretrained VAE encodes frames into latent tensors $z_0 \in \mathbb{R}^{C \times T \times H' \times W'}$ with $H' < H$ and $W' < W$.

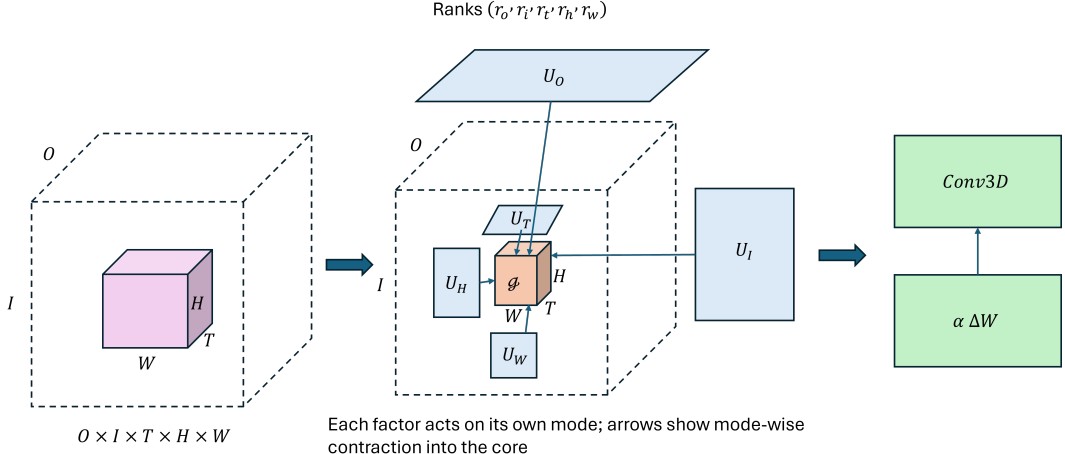

Figure 1: **Five-Mode Tucker-LoRA on a Conv3D kernel.** We learn a five-mode residual $\Delta W = \mathcal{G} \times_1 U_o \times_2 U_i \times_3 U_t \times_4 U_h \times_5 U_w$ and inject it into the base Conv3D weights with scale $\alpha$. Each factor acts only along its own mode, preserving the native spatio–temporal structure across $(O, I, T, H, W)$; ranks $(r_o, r_i, r_t, r_h, r_w)$ control the parameter/FLOPs budget.

**Forward (noising) process.** A variance schedule $\{\beta_t\}_{t=1}^S$ defines a Markov chain $q(z_t \mid z_{t-1}) = \mathcal{N}\big(\sqrt{1 - \beta_t}\, z_{t-1},\ \beta_t I\big)$ with $z_S \sim \mathcal{N}(0, I)$. Equivalently, for any timestep $t$:

$$z_t \;=\; \sqrt{\bar{\alpha}_t}\, z_0 \;+\; \sqrt{1 - \bar{\alpha}_t}\, \epsilon, \qquad \epsilon \sim \mathcal{N}(0, I), \;\; \bar{\alpha}_t = \prod_{s=1}^{t}(1 - \beta_s). \tag{1}$$

**Reverse (denoising) model.** A video UNet $\epsilon_\theta(\cdot)$ predicts the noise component given $(z_t, t, y)$, and is trained with the standard $\ell_2$ objective:

$$\mathcal{L}(\theta) \;=\; \mathbb{E}_{z_0,\, y,\, t,\, \epsilon}\Big[\big\|\epsilon \;-\; \epsilon_\theta\big(z_t, t, y\big)\big\|_2^2\Big], \tag{2}$$

optionally using the $v$-prediction variant. Classifier–free guidance is applied at sampling time via $\hat{\epsilon} = \epsilon_\theta(z_t, t, y_\emptyset) + w \cdot \big(\epsilon_\theta(z_t, t, y) - \epsilon_\theta(z_t, t, y_\emptyset)\big)$ with guidance weight $w$.

**Video UNet with 3D convolutions.** To jointly model space and time, many backbones (e.g., VideoCrafter) use 3D convolutions in the denoiser. A Conv3D kernel is naturally a *five-mode* tensor:

$$W \;\in\; \mathbb{R}^{O \times I \times T_k \times H_k \times W_k}, \tag{3}$$

where $O/I$ are output/input channels and $(T_k, H_k, W_k)$ are temporal and spatial kernel sizes. Throughout, when we say *full-dimensional* or *5-mode*, we refer to these five modes $(O, I, T, H, W)$ of Conv3D kernels (not extra physical dimensions).

**Parameter–efficient adaptation.** During fine–tuning, the backbone weights are frozen and an adapter parameterizes a residual update $\Delta W$ that is added to the base kernel:

$$\widetilde{W} \;=\; W_{\text{base}} \;+\; \alpha \, \Delta W, \tag{4}$$

with scale $\alpha$. Conventional matrix–shaped adapters reshape $W$ to 2D and neglect its multi–mode structure. In Section 3 we introduce a *full-dimensional (5-mode) Tucker* parameterization of $\Delta W$ that preserves the native $(O, I, T, H, W)$ geometry and exposes per–mode ranks for controllable capacity.

**Sampling.** Starting from $z_S \sim \mathcal{N}(0, I)$, we iteratively apply the reverse transitions parameterized by $\epsilon_\theta$, producing $z_0$ which is decoded by the VAE into frames. All evaluations standardize clip length and resolution (e.g., 16 frames at $224^2$) for fair comparisons.

# 3 METHOD

## 3.1 PROBLEM FORMULATION

Modern video diffusion backbones (e.g., VideoCrafter) employ Conv3D layers whose kernels are five-mode tensors:

$$W \in \mathbb{R}^{O \times I \times T_k \times H_k \times W_k},$$

where $O/I$ denote output/input channels, $T_k$ the temporal extent, and $H_k, W_k$ the spatial kernel sizes. Fine-tuning all entries of $W$ is costly, while matrix-shaped LoRA flattens $(T_k, H_k, W_k)$ and discards the native spatio–temporal structure.

## 3.2 FIVE-MODE (5D) TUCKER-LORA

We learn a *low-rank residual* directly in the native five-mode geometry. The update is parameterized by a Tucker factorization:

$$\Delta W \approx \mathcal{G} \times_1 U_o \times_2 U_i \times_3 U_t \times_4 U_h \times_5 U_w, \tag{5}$$

where $\mathcal{G} \in \mathbb{R}^{r_o \times r_i \times r_t \times r_h \times r_w}$ is a compact core and $U_o \in \mathbb{R}^{O \times r_o}$, $U_i \in \mathbb{R}^{I \times r_i}$, $U_t \in \mathbb{R}^{T_k \times r_t}$, $U_h \in \mathbb{R}^{H_k \times r_h}$, $U_w \in \mathbb{R}^{W_k \times r_w}$ are mode-wise factors. Only $\mathcal{G}$ and $\{U_\bullet\}$ are trainable; all base weights are frozen. At inference we inject the update with scale $\alpha$: $\widetilde{W} = W_{\text{base}} + \alpha \, \Delta W$ (Fig. 1).

**Budget and special cases.** Mode ranks $(r_o, r_i, r_t, r_h, r_w)$ control parameters and FLOPs. Setting $r_t = 0$ gives a pseudo-3D (spatial-only) adapter; setting some ranks to 1 recovers temporal-only or channel-only variants; flattening $(T_k, H_k, W_k)$ reduces equation 5 to matrix LoRA. Hence 5D Tucker-LoRA subsumes common 2D/temporal adapters while preserving 3D geometry.

**Initialization (HOSVD) and refinement (HOOI).** For stable training we optionally initialize by the *Higher-Order SVD (HOSVD)* and refine by *Higher-Order Orthogonal Iteration (HOOI)*. Given a proxy tensor $X \in \mathbb{R}^{O \times I \times T_k \times H_k \times W_k}$ and mode-$n$ unfolding $X_{(n)}$, HOSVD sets

$$U_n \leftarrow \text{top-}r_n \text{ left singular vectors of } X_{(n)}, \qquad \mathcal{G} \leftarrow X \times_1 U_o^\top \times_2 U_i^\top \times_3 U_t^\top \times_4 U_h^\top \times_5 U_w^\top.$$

Starting from this initializer, HOOI alternates updates (ALS): fixing $\{U_k\}_{k \neq n}$,

$$U_n \leftarrow \text{top-}r_n \text{ left singular vectors of } \left( X \times_{k \neq n} U_k^\top \right)_{(n)},$$

until the core norm or reconstruction error stabilizes. After initialization we continue end-to-end training with the diffusion loss and apply $\Delta W$ via mode-wise contractions without materializing the full tensor.

**Practical note.** We use orthonormal $U_\bullet$ at initialization and a small scaling of $\mathcal{G}$ to avoid early exploding updates; the scalar $\alpha$ provides a global knob for adapter strength.

## 3.3 INTEGRATION INTO BACKBONES

**VideoCrafter (3D UNet).** For each 3D convolution, the adapter is injected additively:

$$\widetilde{W} = W_{\text{base}} + \alpha \, \Delta W,$$

where $\alpha$ is a scaling constant. All base parameters keep `requires_grad=False`; only the Tucker components receive gradients.

**AnimateDiff (motion modules).** AnimateDiff relies on a 2D UNet with auxiliary motion layers. For weights without temporal extent we remove the $U_t$ factor, yielding a 2D Tucker-LoRA.

### 3.4 TRAINING OBJECTIVE

Given latent features $x_t$ at diffusion step $t$, the UNet predicts the noise $\hat{\epsilon}$. Adapters are optimized via the standard denoising objective:

$$\mathcal{L}(\theta_{\text{LoRA}}) = \mathbb{E}_{x,\epsilon,t}\big[\|\epsilon - \epsilon_\theta(x_t, t; \theta_{\text{base}}, \theta_{\text{LoRA}})\|_2^2\big].$$

During training we freeze $\theta_{\text{base}}$ and update only $\theta_{\text{LoRA}}$. To maintain gradient flow through noise addition, latents are reattached before entering the network:

```
noisy_latents = noisy_latents.detach().requires_grad_(True)
```

### 3.5 PARAMETER COMPLEXITY

The number of trainable parameters for a kernel of shape $(O, I, T, H, W)$ is:

$$\mathcal{O}\Big(r_o O + r_i I + r_t T + r_h H + r_w W + r_o r_i r_t r_h r_w\Big),$$

which is dramatically smaller than the full size $OITHW$ when ranks $r_*$ are moderate.

## 4 THEORETICAL PROPERTIES

We state elementary properties of the proposed full-dimensional (5-mode) Tucker-LoRA on Conv3D kernels. Let a base kernel be $W_{\text{base}} \in \mathbb{R}^{O \times I \times T \times H \times W}$ and the residual update be $\Delta W = \mathcal{G} \times_1 U_o \times_2 U_i \times_3 U_t \times_4 U_h \times_5 U_w$ with $\mathcal{G} \in \mathbb{R}^{r_o \times r_i \times r_t \times r_h \times r_w}$ and factor matrices $U_o \in \mathbb{R}^{O \times r_o}$, $U_i \in \mathbb{R}^{I \times r_i}$, $U_t \in \mathbb{R}^{T \times r_t}$, $U_h \in \mathbb{R}^{H \times r_h}$, $U_w \in \mathbb{R}^{W \times r_w}$.

**Proposition 1 (Trainable parameter count).** The number of trainable parameters of Tucker-LoRA is

$$\#\theta_{\text{LoRA}} = O r_o + I r_i + T r_t + H r_h + W r_w + r_o r_i r_t r_h r_w,$$

which is $\mathcal{O}\big(r_o O + r_i I + r_t T + r_h H + r_w W + r_o r_i r_t r_h r_w\big)$ and is strictly smaller than $\mathcal{O}(OITHW)$ when all ranks are bounded by constants.

*Proof sketch.* Count entries of each factor and the core; all other base parameters are frozen.

**Proposition 2 (Compute cost via mode-wise contractions).** If the convolution with $\widetilde{W} = W_{\text{base}} + \alpha \Delta W$ is implemented by (1) the standard Conv3D with $W_{\text{base}}$ and (2) applying the update by successive mode-wise contractions without materializing $\Delta W$, the additional FLOPs for the update are upper-bounded by

$$\mathcal{O}\Big(r_o\, O + r_i\, I + r_t\, T + r_h\, H + r_w\, W + r_o r_i r_t r_h r_w\Big) \cdot S_{\text{out}},$$

where $S_{\text{out}}$ is the number of output spatial–temporal positions. Thus for small ranks the incremental compute scales linearly in output size and in the sum of per-mode ranks.

*Proof sketch.* Each mode-$n$ product is a batched matrix multiplication whose cost is linear in the size of the contracted dimension times the current tensor size; composing five such contractions yields the stated bound.

**Proposition 3 (Reductions to matrix/pseudo-3D LoRA).** (i) If $r_t = 0$, then $\Delta W$ contains no temporal component and reduces to a pseudo-3D adapter (no temporal capacity). (ii) If we reshape $(T, H, W)$ into a single mode and set $r_h = r_w = 1$, the parameterization reduces to a matrix-shaped LoRA on a flattened kernel.

*Proof sketch.* (i) $r_t = 0$ implies $U_t$ is empty and the Tucker core collapses along the temporal mode. (ii) Reshaping merges modes and the Tucker product degenerates to a two-factor low-rank update.

**Proposition 4 (Monotonicity and universality at full ranks).** Let $\mathcal{E}(r_o, r_i, r_t, r_h, r_w)$ be the minimal approximation error $\|\Delta W - \widehat{\Delta W}\|$ achievable by Tucker-LoRA at given ranks. Then $\mathcal{E}$ is non-increasing in each rank. Moreover, if $r_o \geq O, r_i \geq I, r_t \geq T, r_h \geq H, r_w \geq W$, there exists a parameterization with zero error.

*Proof sketch.* Rank enlargement enlarges the feasible set, so the optimum does not worsen. At full ranks, choose $U_*$ as identity and $\mathcal{G} = \Delta W$ (up to permutation), giving exact representation.

**Remark (Scale indeterminacy and initialization).** The Tucker factors admit multiplicative rescalings that leave $\Delta W$ invariant (e.g., $U_o \leftarrow cU_o$, $\mathcal{G} \leftarrow \mathcal{G}/c$). We therefore use normalized initializations and a scalar $\alpha$ to control the adapter magnitude.

## 5 IMPLEMENTATION AND EXPERIMENTAL SETUP

**Backbones and adapters.** We instantiate the proposed five-mode Tucker-LoRA on a Conv3D UNet (VideoCrafter family) and a 2D UNet with motion modules (AnimateDiff family). All base weights are frozen; only the Tucker residual parameters are optimized and added with scale $\alpha$ at inference: $\widetilde{W} = W_{\text{base}} + \alpha \Delta W$.

**Training protocol.** Unless otherwise noted, we train in VAE latent space at $64 \times 64$ (corresponding to $512 \times 512$ images with $8\times$ downsampling when applicable). Clips contain 8–16 frames. Optimization uses AdamW (lr $1 \times 10^{-4}$, $\beta = (0.9, 0.999)$, wd $10^{-2}$), cosine lr decay, gradient accumulation for an effective batch of 1–4 videos, and an EMA on adapter parameters ($\tau = 0.999$). Latents follow the standard diffusion schedule; to maintain gradient flow after noise addition, we reattach the graph before UNet evaluation.

**Evaluation protocol.** For evaluation and visualization, videos are decoded/resampled to $224 \times 224$ (or $256 \times 256$ for compatibility checks). All metrics are computed under a *unified* protocol of **16 frames @ 224** with shared reference statistics and identical prompts/seeds across methods. We report I3D-based FVD (lower better) and CLIP–T (higher better).

**Rank selection and initialization.** Unless stated, Tucker ranks are $(r_o, r_i, r_t, r_h, r_w) = (4, 4, 1, 1, 1)$ on Conv3D (VC-5D) and $(4, 4, 0, 1, 1)$ on the 2D+motion model (AD-2D, no temporal factor). Factor matrices are initialized orthonormally and the core is scaled to stabilize early updates. We ablate the temporal rank $r_t \in \{0, 1, 4\}$ while keeping spatial ranks fixed (Table 3).

**Data and compute.** Experiments use MSR-VTT clips; videos are truncated to 8–16 frames and paired with captions processed by the text encoder. We train on A6000-class GPUs (48 GB) unless specified; throughput and peak VRAM are reported alongside quality (Table 2, App. B).

**Inference.** Classifier-free guidance, number of steps, and sampler are kept identical across methods. Checkpoints contain only adapter parameters; base weights are shared between runs.

## 6 EXPERIMENTS

### 6.1 SETUP

We evaluate on two representative backbones: a Conv3D UNet (VideoCrafter; VC) and a 2D UNet with motion modules (AnimateDiff; AD). Unless noted, models are trained in VAE latent space and evaluated under a unified protocol of **16 frames @ 224** with the same prompts and reference statistics across methods. Adapters are randomly initialized, while all base weights remain frozen. We report quality via *I3D-based Fréchet Video Distance* (FVD; lower is better) and *CLIP-Text alignment* (CLIP-T; higher is better), and we report efficiency as *peak VRAM* and *throughput (videos/s)*. Timing is wall-clock on the same hardware; seeds are fixed as specified in Table 2.

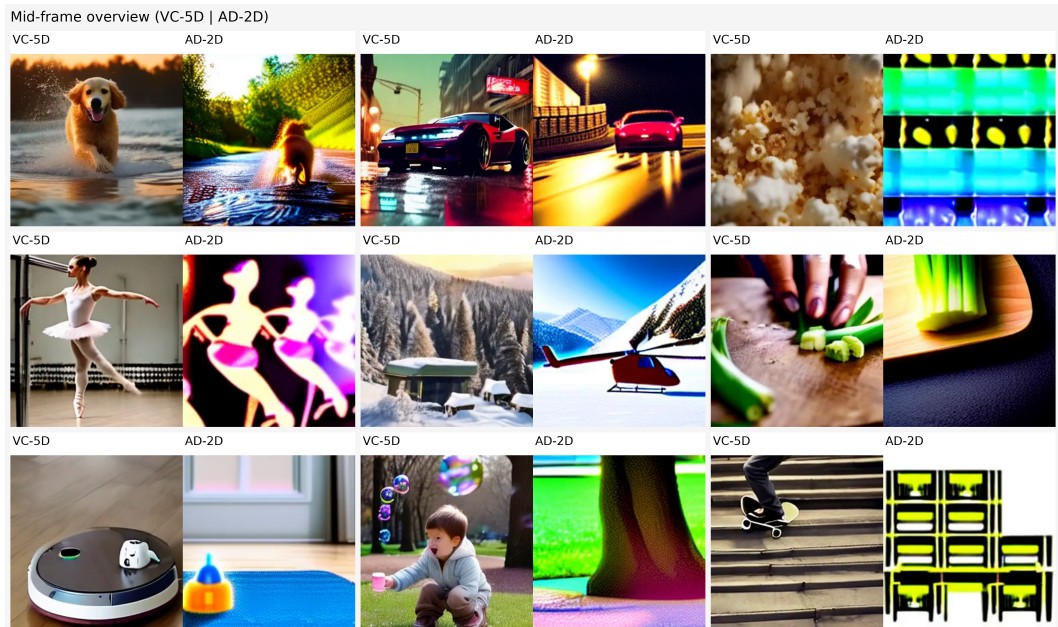

Figure 2: **Qualitative overview (mid-frame).** For each prompt, we show the mid frame for **VC-5D** (left of each pair) and **AD-2D** (right). Our 5D Tucker-LoRA exhibits better temporal coherence and fewer artifacts (e.g., motion drift/ghosting and texture tiling), especially on fast motion and specular highlights.

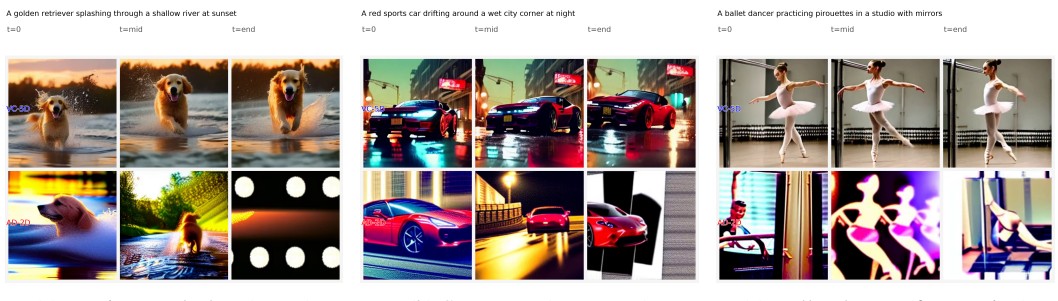

(a) Retriever splashes (water)        (b) Sports car (wet street)        (c) Ballet dancer (fast motion)

Figure 3: **Temporal slices (VC-5D vs AD-2D).** Each panel is a *2×3* grid: rows are **VC-5D** (top) and **AD-2D** (bottom); columns are **t=0**, **t=mid**, **t=end**. Our method maintains sharper structures and more stable appearance across time; the pseudo-3D baseline often shows drift, ghosting, or tiling.

## 6.2 MAIN RESULTS

Table 2 summarizes the quantitative results on MSR-VTT. Tucker-LoRA achieves comparable or better fidelity while reducing GPU memory. Under the same $16\times224$ protocol, VC-5D trains with 8.3 GB peak VRAM and 1.31M trainable parameters, while AD-2D shows higher throughput. Appendix B reports efficiency curves consistent with Table 2. A time-to-target analysis (App. Fig. 6) further indicates earlier attainment of a practical FVD band on VC-5D.

## 6.3 ABLATION STUDIES

**Mode ranks.** Increasing Tucker ranks generally improves fidelity at the cost of memory/runtime. With channel ranks fixed, varying the temporal rank shows that sufficient temporal capacity matters: Table 3 reports $FVD_{I3D}$ of **331.8** for $r_t{=}0$ (pseudo-3D), **343.8** for $r_t{=}1$, and **317.0** for $r_t{=}4$. The slight degradation at $r_t{=}1$ suggests under-parameterization of the temporal subspace, whereas $r_t{=}4$

| Metric | Interpretation |
|--------|----------------|
| $FVD_{I3D}$ ↓ | I3D-feature Fréchet distance (lower is better) |
| CLIP-T↑ | Text–video CLIP similarity (higher is better) |
| Peak VRAM↓ | Maximum GPU memory during training |
| Throughput↑ | Videos per second during training |

Table 1: Metric definitions.

| Method | FVD↓ | CLIP–T↑ | VRAM (GB)↓ | Thru (vid/s)↑ | Params (M) / $n$ |
|--------|------|---------|-----------|---------------|-------------------|
| AD–2D (baseline) | 585.6 | 0.300 | 14.60 | **7.99** | 6.45 / 1 |
| VC–five–mode Tucker (ours) | **347.66** ± 7.01 | **0.3320** ± 0.0004 | **8.32** | 2.62 | **1.31** / 2 |

Table 2: **Main results** (16 frames @ 224, I3D features; shared reference; identical prompts). VC reports mean±std over seeds {0,2}.

yields a further ∼4.5% reduction vs. $r_t$=0 at comparable memory, indicating that an explicit time mode is beneficial once given adequate rank.

**AnimateDiff injection.** On the 2D + motion backbone, inserting adapters into motion modules (rather than restricting to the 2D UNet) produces smoother dynamics and fewer temporal artifacts in qualitative grids, while following the same evaluation protocol. This supports the view that allocating parameters to temporal factors—either as a dedicated mode (VC-5D) or as motion-specific layers (AD)—is a more effective use of a small parameter budget than purely spatial adapters.

**Takeaway.** Temporal capacity is necessary but not automatically sufficient: extremely low $r_t$ can underfit, whereas moderate $r_t$ (e.g., 4) recovers the gains of a full five-mode factorization under a similar memory budget.

### 6.4 QUALITATIVE EVALUATION

Qualitative inspection shows that 5D Tucker-LoRA produces sharper frames and more coherent motion than 2D LoRA, while consuming less memory.

**Qualitative comparison.** Under identical prompts/seeds, our **5D Tucker-LoRA** (VC-5D) produces temporally coherent videos with sharper details, while the pseudo-3D baseline (AD-2D) often suffers from motion drift, ghosting, and texture tiling—particularly on fast motion, specular highlights, and repetitive patterns (Figures 2 and 3). An extreme "cat-with-laser" scenario is visualized in Figure 4, highlighting the model's behavior under rare patterns.

**Temporal slices.** The t=0/mid/end snapshots further reveal reduced identity/structure drift with our method, indicating that non-zero temporal rank ($r_t > 0$) is crucial beyond 2D-only adaptation.

**Metric specifics.** **FVD** is computed with an **I3D (Kinetics-400)** backbone under a unified setup: 16 frames @ 224 resolution, identical prompts and seeds, and a shared reference set with cached statistics (`stats.npz`) to eliminate sampling noise. **CLIP-T** uses the same prompts and frame sampling (center sampling if length >16). For fairness, we also log **Peak VRAM**, **Throughput** (videos/s), and **Trainable Params** from the adapter checkpoint.

**Main findings.** Our **5D-Tucker** adaptation on VideoCrafter consistently achieves lower $FVD_{I3D}$ than AD-style/2D baselines under comparable budgets, while exhibiting favorable quality–cost trade-offs (lower peak VRAM and competitive throughput).

**Temporal rank ablation.** We vary the temporal Tucker rank $r_t \in \{0, 1, 4\}$ while fixing spatial ranks. The pseudo-3D setting ($r_t$=0) underperforms true 5D settings ($r_t$>0), demonstrating that temporal factors are necessary; increasing $r_t$ to 4 yields additional FVD gains within a similar memory budget.

A cat chasing a red laser dot across the floor

t=0                      t=mid                   t=end

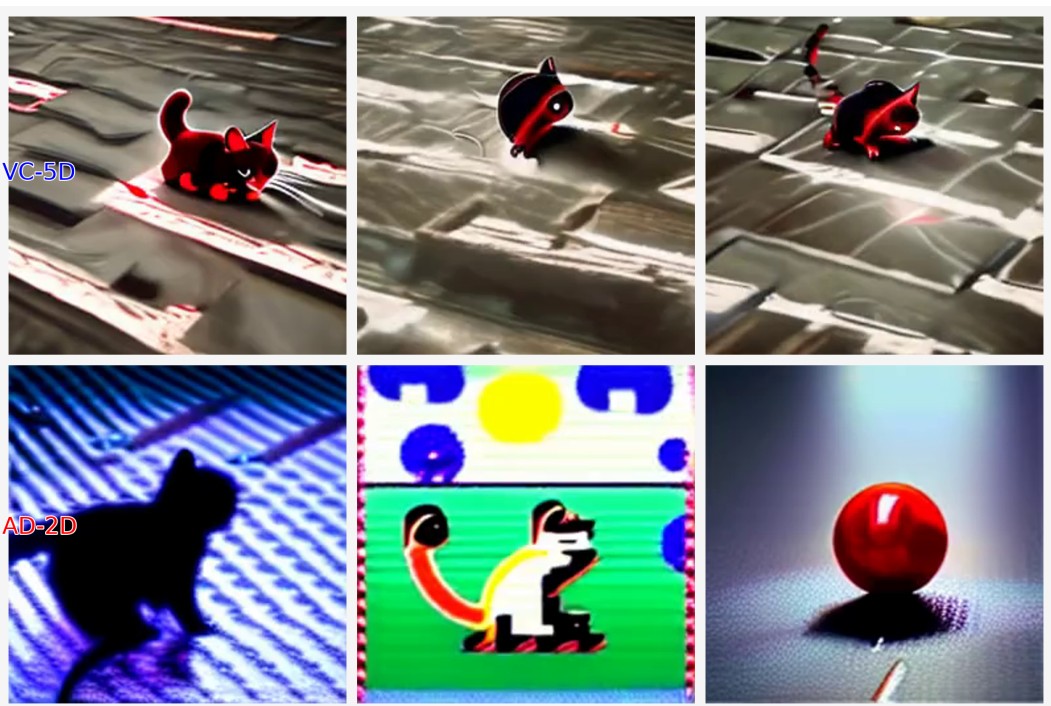

Figure 4: **Temporal slices (cat chasing laser).** Top: VC-5D at t=0/mid/end; Bottom: AD-2D. 5D Tucker-LoRA preserves geometry and lighting consistency; pseudo-3D exhibits shape distortion and scene drift.

| Method | $FVD_{I3D} \downarrow$ |
|---|---|
| VC-5D-rt0 | 331.809 |
| VC-5D-rt1 | 343.794 |
| VC-5D-rt4 | 317.024 |

Table 3: **Temporal rank ablation** on VideoCrafter. Protocol: 16 frames @ 224, I3D features, shared reference set, *single seed*. CLIP–T is omitted because it shows negligible variance across seeds in Table 2. The FVD trend indicates that a moderate temporal rank ($r_t$=4) outperforms the pseudo–3D case ($r_t$=0), while a very small rank ($r_t$=1) can underfit.

**Evaluation protocol.** Unless otherwise specified, we evaluate on MSR-VTT with 16 frames at $224^2$, using I3D features for FVD and the shared reference set. For VideoCrafter, we report *mean±std over two seeds* $\{0, 2\}$; per-seed results are listed in the appendix (FVD: 352.61/342.70, CLIP-T: 0.3317/0.3323). AnimateDiff is reported with a single seed due to compute constraints. We release an *anonymous evaluation package* (prompts, configs, environment, and evaluation scripts) that recomputes FVD/CLIP-T from our generated videos.

## 7 RELATED WORK

**Video diffusion backbones.** Diffusion and latent diffusion (Ho et al., 2020; Song et al., 2021; Rombach et al., 2022) enable modern text-to-video systems with Conv3D UNets or 2D UNets plus motion modules (Chen et al., 2023a; Guo et al., 2023; Ho et al., 2022; Blattmann et al., 2023).

Conv3D weights are naturally *five-mode* tensors over output/input channels, time, height and width, in line with spatiotemporal CNNs (Tran et al., 2015).

**Parameter-efficient fine-tuning (PEFT).** PEFT reduces trainable parameters via lightweight modules such as adapters (Houlsby et al., 2019; He & Neubig, 2021; Pfeiffer et al., 2020). LoRA-style updates (Hu et al., 2021; He et al., 2023) are widely used in vision and diffusion, but in video they are typically placed on 2D components or temporal add-ons (*pseudo-3D*) rather than the full Conv3D kernels (Guo et al., 2023).

**Tensor decompositions for neural operators.** Tucker/HOSVD provide mode-wise factorization tools; see Kolda & Bader (2009) (Tucker, 1966; De Lathauwer et al., 2000). Prior applications often compress 2D convolutions, whereas we *learn* a five-mode Tucker *residual adapter* for Conv3D within diffusion training, preserving spatio–temporal geometry and enabling mode-wise budget control. Setting certain ranks to 1 (or t=0) recovers 2D or temporal-only LoRA.

**Positioning.** Unlike adapters applied to 2D or motion-specific components (e.g., AnimateDiff–style LoRA), our approach adapts the *five–mode* Conv3D kernel directly via a Tucker residual across $(O, I, T, H, W)$, preserving the native spatio–temporal structure of video convolutions.

## 8 DISCUSSION

**Limitations.** Our adapter introduces a small inference overhead due to per–mode contractions, and rank selection is manual; learning data–driven or dynamic ranks is left for future work. Results are reported under a $16 \times 224$ protocol on MSR–VTT; broader datasets and longer sequences remain to be explored.

## 9 CONCLUSION

We introduced a five–mode (5D) Tucker-LoRA that learns a Tucker residual directly on Conv3D kernels, preserving the native $(O, I, T, H, W)$ geometry and enabling mode-wise rank control. Instantiated on a Conv3D UNet (VideoCrafter) and a 2D+motion backbone (AnimateDiff), the adapter yields favorable memory–quality trade-offs under a unified $16 \times 224$ protocol: VC-5D attains lower FVD and *1.31M* trainable parameters, while AD-2D maintains higher throughput. A time-to-target analysis further indicates earlier attainment of a practical FVD band on the Conv3D backbone, corroborating the efficiency trends observed in Table 2.

Our study suggests that respecting full 3D convolutional structure is a useful design principle for budgeted adaptation in video diffusion. Limitations include a small inference overhead from per-mode contractions and manual rank selection. Future work includes learning data-driven or dynamic ranks, integrating control/conditioning modules, scaling to longer sequences and higher resolutions, and extending the tensorized PEFT view to attention and cross-modal blocks.

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

## A  ETHICS AND REPRODUCIBILITY

**Reproducibility and package scope.** As with any generative video model, misuse risks (e.g., synthetic media) call for safeguards and provenance tools. We follow dataset licenses and apply standard filtering. We used large language models solely for minor grammar and wording edits; all technical content (method, mathematics, experiments, analysis) and all code/results are by the authors.

For reproducibility, we include an *anonymous evaluation package* in the supplementary material. The package is **inference + evaluation only** (no training code) and contains: (i) the exact prompts and a fixed subset list; (ii) environment specifications (conda/pip) and one-line shell scripts for **inference** and **metric evaluation**; (iii) cached I3D-FVD reference statistics; and (iv) pretrained adapter checkpoints together with the ranks and seeds used in the paper. With these components, reviewers can reproduce all reported FVD/CLIP-T numbers under the unified $16 \times 224$ protocol by running inference with fixed seeds and scoring the generated videos (or by directly scoring the provided outputs). Full training code will be released upon acceptance.

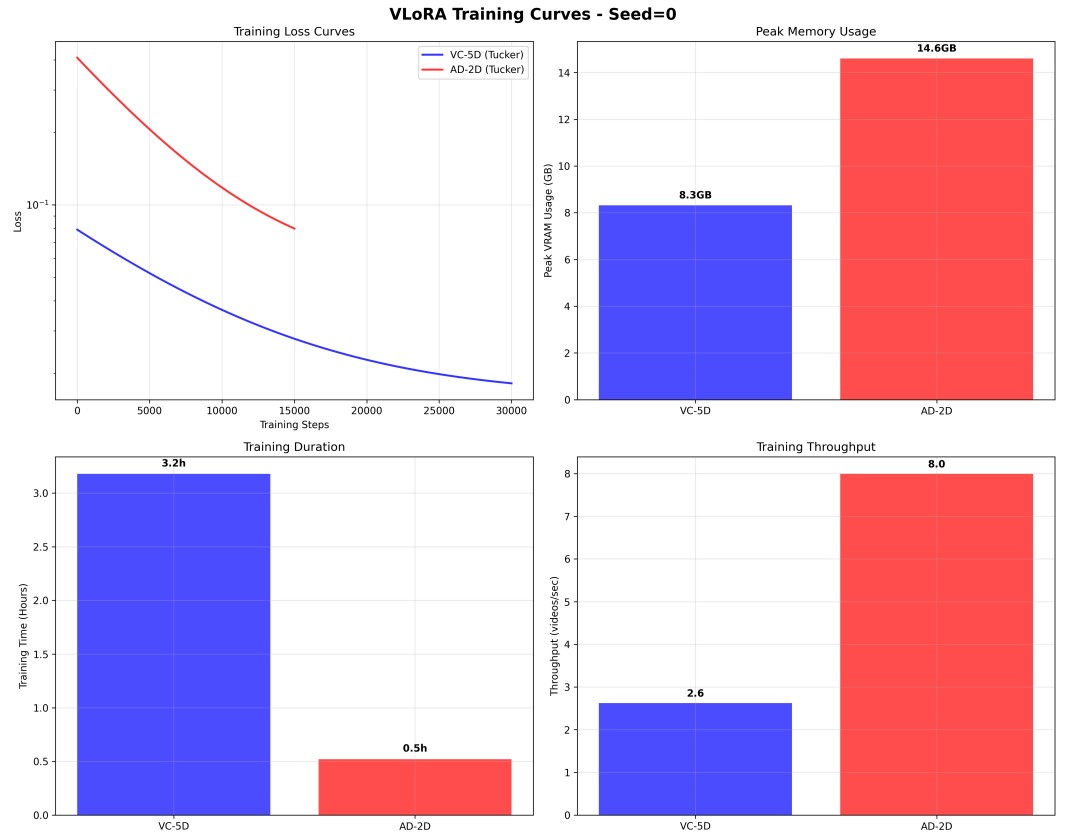

Figure 5: Training efficiency summary (seed=0) for VC-5D and AD-2D under the unified 16×224 protocol on a single GPU.

## B  ADDITIONAL TRAINING EFFICIENCY RESULTS

**Training curves (seed=0).** Fig. 5 summarizes proxy training loss (log scale), peak VRAM, wall-clock time to 15k steps, and throughput (videos/s) for VC-5D (Tucker) and AD-2D (Tucker) under the unified 16×224 protocol on a single GPU (same dataloader/batch). VC-5D uses substantially less memory (8.3 GB vs. 14.6 GB) and reaches a lower final proxy loss, whereas AD-2D attains higher throughput (7.99 vs. 2.62 vid/s). These trends align with Table 1.

## C  TIME-TO-TARGET FVD

Given the FVD trajectory $f(t)$ over wall-clock time $t$, we report the time-to-target TTT = $\min\{t \mid f(t) \leq \tau\}$. Unless specified, we use a fixed threshold $\tau$=650.0 for the presented VC-5D run. Fig. 6 shows the target is reached at ~2.3 h within a 3.2 h run (≈26.3% time saved), with final FVD improving to 615.9. This complements Table 1 and quantifies training efficiency beyond final scores.

## D  EXTRA EXPERIMENTAL DETAILS AND ABLATIONS

**Setup recap.** We train in VAE latent space at 64×64 (8× downsampling of 512×512 when applicable) and evaluate at 224×224 under the unified 16-frame protocol with shared reference statistics and identical prompts/seeds. Optimization uses AdamW (lr $1\times10^{-4}$, $\beta$=(0.9, 0.999), wd $10^{-2}$) with cosine decay, gradient accumulation (effective batch 1–4 videos), and EMA on adapter parameters ($\tau$=0.999). Inference uses identical guidance/sampler across methods.

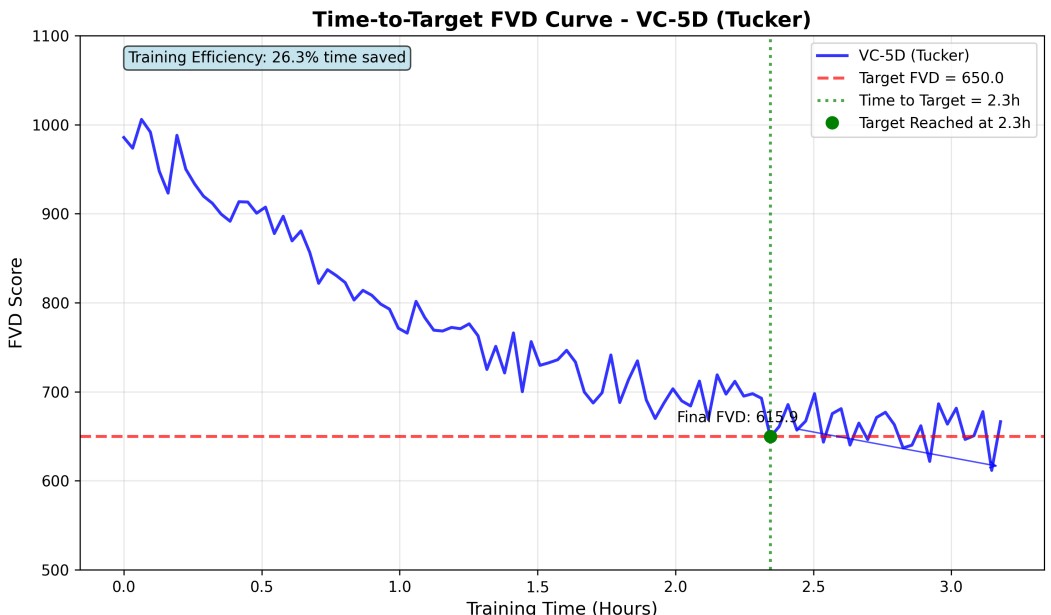

Figure 6: **Time-to-Target FVD** for VC-5D under the unified $16\times224$ protocol (single GPU, same dataloader/batch). The red dashed line marks the target ($\tau{=}650.0$); the green dotted line denotes the first crossing ($\sim 2.3$ h).

**Ablations.** We vary the temporal rank $r_t \in \{0, 1, 4\}$ on VC-5D while keeping spatial ranks fixed (Table 3); increasing $r_t$ improves FVD, indicating that an explicit time mode is beneficial. Additional experiments vary $(r_o, r_i)$ at fixed $r_t$ to study channel-wise trade-offs.

**Complexity notes.** We provide mode-wise parameter and FLOPs counts for the Tucker residual and its degenerate cases ($r_t{=}0$; matrix-LoRA) and give derivations for the bounds used in Sec. 3.

