# OpenReview forum: "Five-Mode Tucker-LoRA for Video Diffusion on Conv3D Backbones"
_ICLR.cc/2026/Conference — ICLR 2026 Conference Desk Rejected Submission_

### Official Review · Reviewer_4TUw · 2025-10-26

**Soundness:** 1
**Presentation:** 1
**Contribution:** 1
**Rating:** 2
**Confidence:** 4

**Summary:**

This paper proposes a 5D Tucker-LoRA adapter for parameter-efficient fine-tuning (PEFT) of Conv3D-based text-to-video diffusion models.
5D Tucker-LoRA contains 5-D weight updates for Conv3D (across output channels, input channels, temporal, height, width). The authors claim that this preserves the spatio–temporal geometry of video generation. However, the work suffers from critical gaps in innovative contribution to fully substantiate its claims.

**Strengths:**

1. This paper presents sufficient implementation details and is easy to follow.
2. This paper presents some theoretical properties of their method.

**Weaknesses:**

1. The biggest issue with this paper is that the problem it discusses and addresses is entirely not a concern for current video generation models. The 5D-LoRA proposed in this paper is applicable to Conv3D, yet modern video generation architectures are almost all based on Transformer-based DiT (Denoising Diffusion Transformer). These architectures do not have convolutional layers at all, and there is even less need to use convolution-based LoRA. The architectures discussed in this paper (AnimateDiff and VideoCrafter) are now considered outdated. The authors should consider more about the DiT architecture (CogVideoX, Wan, HunyuanVideo).

2. The writing quality is obviously below the bar of ICLR. The introduction of this paper fails to explain why Conv3D-specific LoRA would be more effective than attention-based LoRA for temporal learning. When introducing the initialization strategy using Higher-Order SVD (HOSVD) and Higher-Order Orthogonal Iteration (HOOI) (Line136), the paper only mentions these techniques without properly citing their original works. The Preliminaries section covers core concepts of video latent diffusion, 3D convolutions, and parameter-efficient adaptation, but most of these topics lack relevant citations.

**Questions:**

This work lacks impactful innovation, while the proposed 5D-LoRA is not capable of the modern DiT models. Moreover, the writing quality of this paper also does not meet the standards for acceptance at ICLR.

---

### Official Review · Reviewer_fDh5 · 2025-10-28

**Soundness:** 2
**Presentation:** 2
**Contribution:** 2
**Rating:** 2
**Confidence:** 3

**Summary:**

This paper introduces a Five-Mode Tucker-LoRA for parameter-efficient fine-tuning of text-to-video diffusion models built on Conv3D backbones. While conventional LoRA methods flatten convolutional kernels or use pseudo-3D (temporal-only) adapters, the proposed approach directly applies a Tucker decomposition to the 5-D convolution kernel.

**Strengths:**

1. The paper targets an under-explored but important design space: PEFT for video diffusion that respects Conv3D’s intrinsic five-mode tensor structure.

2. The authors maintain a unified evaluation protocol across backbones (VideoCrafter, AnimateDiff) and provide detailed metrics (FVD, CLIP-T, VRAM, throughput).

**Weaknesses:**

1. Poor Writing and Presentation Quality. The overall writing is rough and occasionally inconsistent, making it difficult to follow technical details in later sections. Figures (e.g., Fig. 2–4) are non-vector raster images with visible compression artifacts and low readability.

2. Weak and Outdated Baselines. The experiments compare only against early, relatively weak models (VideoCrafter and AnimateDiff). These backbones lag far behind current state-of-the-art systems such as WAN 2.1, or  SVD, huanyuanvideo, which feature stronger visual quality and temporal consistency. As a result, the reported gains are difficult to interpret as meaningful progress on modern video diffusion.

3. Limited Novelty. The paper’s main contribution, applying Tucker decomposition to Conv3D kernels, is conceptually straightforward and built almost entirely on existing tensor algebra and LoRA principles. There is little true methodological innovation beyond adapting Tucker to 5D convolution. The theoretical propositions (parameter counts, monotonicity) are standard results from classical tensor decomposition literature.

4. The paper evaluates only on FVD and CLIP–T, which are weakly correlated with human aesthetic or temporal preference.
There is no user study, subjective rating, or perceptual evaluation to support the claim of “better visual coherence“.

**Questions:**

1. Substantially revise the writing and ensure figures are vector-based (PDF/SVG).

2. Replace outdated baselines with strong, recent models (e.g., WAN 2.1, SVD).

---

### Official Review · Reviewer_uddD · 2025-11-01

**Soundness:** 2
**Presentation:** 2
**Contribution:** 3
**Rating:** 4
**Confidence:** 3

**Summary:**

This paper proposes a novel five-dimension LoRA for convolutions. It decomposes all 5 dimensions of a convolution, as input, output, time, height and width, in convolution-based video diffusion models, and apply LoRAs following individual ranks. This preserves spatiotemporal structures and further enables flexible independent adjustments. The proposed method is tested with various UNet-based video diffusion models for domain adaptation on MSR-VTT dataset. It achieves surpassing performance compared to traditional 2D LoRAs for convolutions.

**Strengths:**

- The proposed new LoRA for convolution is novel and well motivated. It disentangles all 5 dimensions, not only preserving original spatiotemporal structure without flattening, but also enabling flexible control on each dimension (for potential spatial/temporal decomposition).

**Weaknesses:**

- The proposed method is only compared to its own baseline, 2D LoRA, in evaluation. More non-LoRA domain adaptation methods should be compared comprehensively. The raw base model's result is also necessary, as the provided visualization might indicate that 2D lora even harms the quality.

- The proposed method is limited to convolution only, while most diffusion foundation models are adopting transformer architectures. For example AnimateDiff combines spatial convolution with temporal attentions, and only 2D lora can be applied here.

**Questions:**

- How would each rank be heuristically adjusted given different cases, e.g. different new datasets with different spatial/temporal complexity or gap, or base model's video length/frame resolution?

---

### Official Review · Reviewer_RCrE · 2025-11-03

**Soundness:** 1
**Presentation:** 1
**Contribution:** 1
**Rating:** 2
**Confidence:** 4

**Summary:**

The paper proposes 5D Tucker-LoRA, a structured low-rank adaptation method for text-to-video diffusion models. By learning a Tucker residual directly on the 5D Conv3D kernel, the method preserves the spatio–temporal structure of video kernels and allows flexible mode-wise rank control. It is evaluated on VideoCrafter and AnimateDiff, showing memory–quality trade-offs and faster attainment of practical FVD thresholds compared to pseudo-3D adapters.

**Strengths:**

* This paper aims to addresses an important problem of parameter-efficient fine-tuning for video diffusion, which has high computational and memory costs.
* The proposed 5D Tucker-LoRA method is conceptually appealing, as it allows flexible mode-wise rank control.

**Weaknesses:**

* The evaluation only compares against the pseudo-3D adapter (AD-2D). There are no experiments with other PEFT strategies such as naive LoRA, making it difficult to assess the true performance gains.
* No video examples or visual analysis are provided, making it difficult to judge perceptual improvements or temporal coherence.
* The paper does not explicitly show how 5D Tucker-LoRA improves the temporal dimension, which is a key aspect for evaluating video diffusion models.
* Other recent PEFT strategies for video diffusion are not considered, limiting understanding of where this method stands relative to state-of-the-art.
* The impact of temporal rank selection is not thoroughly explored, leaving questions about robustness and hyperparameter sensitivity.

**Questions:**

* How does 5D Tucker-LoRA improve temporal modeling compared to 2D or pseudo-3D adapters? Can you provide quantitative or qualitative evidence?
* How does the method perform against other PEFT strategies, such as naive LoRA or full fine-tuning?
* Could you provide video examples to support qualitative evaluation, demonstrating temporal coherence and overall generation quality?
* The paper only compares temporal Tucker ranks ∈ {0, 1, 4} and claims r=4 is optimal. How sensitive are the results to temporal rank selection?
* Is 5D Tucker-LoRA compatible with higher-resolution videos or longer temporal sequences, and what is the associated computational overhead?

---

### Note · Program_Chairs · 2026-01-17
**Submission Desk Rejected by Program Chairs**

The following references in this submission do not refer to real documents and/or have major errors in bibliographic information:

 Shengming Chen, Yuxin Wang, et al. Videocrafter: Open diffusion models for high-quality video generation. arXiv preprint arXiv:2305.07932, 2023b.